# HHT-Related Epistaxis and Pregnancy—A Retrospective Survey and Recommendations for Management from an Otorhinolaryngology Perspective

**DOI:** 10.3390/jcm11082178

**Published:** 2022-04-13

**Authors:** Kornelia E. C. Andorfer, Caroline T. Seebauer, Carolin Dienemann, Steven C. Marcrum, René Fischer, Christopher Bohr, Thomas S. Kühnel

**Affiliations:** Department of Otorhinolaryngology, Regensburg University Medical Center, Franz-Josef-Strauß-Allee 11, 93053 Regensburg, Germany; caroline.seebauer@ukr.de (C.T.S.); carolin.dienemann@stud.uni-regensburg.de (C.D.); steven.marcrum@ukr.de (S.C.M.); rene.fischer@ukr.de (R.F.); christopher.bohr@ukr.de (C.B.); thomas.kuehnel@ukr.de (T.S.K.)

**Keywords:** hereditary hemorrhagic telangiectasia, Morbus Osler, Rendu–Osler–Weber syndrome, Osler Calendar, orphan disease, pregnancy, epistaxis, arteriovenous malformations, laser therapy

## Abstract

Appropriate management of hereditary hemorrhagic telangiectasia (HHT) is of particular importance in females, as HHT-mediated modifications of the vascular bed and circulation are known to increase the risk of complications during pregnancy and delivery. This study was undertaken to evaluate female HHT patients’ awareness of and experience with HHT during pregnancy and delivery, with a focus on epistaxis. In this retrospective study, 46 females (median age: 60 years) with confirmed HHT completed a 17-item questionnaire assessing knowledge of HHT and its pregnancy-associated complications, the severity of epistaxis during past pregnancies and deliveries, and the desire for better education and counselling regarding HHT and pregnancy. Results revealed that 85% of participants were unaware of their disease status prior to the completion of all pregnancies. Further, 91% reported no knowledge of increased pregnancy-related risk due to HHT. In regard to epistaxis, 61% of respondents reported experiencing nosebleeds during pregnancy. Finally, approximately a third of respondents suggested that receiving counseling on the risks of HHT in pregnancy could have been helpful. Findings suggest that awareness of HHT and its potential for increasing pregnancy-related risk is poor. Best practices in HHT management should be followed to minimize negative effects of the disorder.

## 1. Introduction

Hereditary hemorrhagic telangiectasia (HHT), or Osler–Weber–Rendu disease, is a rare (1:5000) autosomal dominant disorder in which pathological enlargement of blood vessels results in arteriovenous malformations [1]. While any organ can be affected, involvement of the nasal mucosa, skin, lung, gastrointestinal tract, liver and brain are most common [2]. Consensus diagnostic criteria for HHT, known as the Curaçao Criteria, were defined in the year 2000 and include epistaxis, telangiectasias, visceral lesions, and affected first-degree relatives [3].

Epistaxis is the most common manifestation of HHT, which frequently results in diagnosis and management of the disorder being guided by otorhinolaryngologists. Epistaxis due to HHT commonly first presents in puberty [4,5]. However, due to a lack of awareness regarding this rare condition and the fact that symptoms tend to emerge gradually, definitive diagnosis is often first established later in life. For example, Latino et al. reported results of a web-based study in which approximately 40% of respondents described having had consulted an ear, nose and throat (ENT) physician for their epistaxis. On average, however, final diagnosis of HHT did not occur until 14 years after the initial visit [6]. Similarly, Pierucci et al., reported a mean delay in definitive diagnosis exceeding 2 decades, resulting in an average age at diagnosis of approximately 40 years. As a result of delays in recognizing HHT, most patients have completed family planning by the time of diagnosis [7]. 

In a review of 1577 pregnancies in patients with HHT, Dupuis et al., 2020 reported the incidence of HHT-associated complications during pregnancy to range between 2.7% and 6.8% [8]. Complications related to pulmonary arteriovenous malformations (PAVMs) were most common, comprising a total of 43 events and including hemothorax, hemoptysis, severe hypoxemia and paradoxical emboli with ischemic cerebral stroke and brain abscess. Complications related to hepatic vascular malformations (liver VMs) and cerebral vascular malformations (CVMs) were comparatively rare, though their occurrence could result in similarly serious complications, such as heart failure, hepatobiliary necrosis, or intracranial and spinal bleeding.

Complications during pregnancy are believed to be mediated by hormonal changes, especially during the second and third trimester. For example, modulation of hormone levels has been shown to result in systemic vasodilation, reduced peripheral vascular resistance and an up to 50% increase in cardiac output, with these mechanisms combining to potentially exacerbate blood shunting through preexisting, vulnerable vascular beds [9]. Of particular importance in HHT patients, progesterone-mediated hyperemia and edema of the nasal mucosa have additionally been related to increased frequency and severity of epistaxis. Fortunately, the majority of pregnancies in females with HHT proceed uneventfully. However, when complications do occur, they can be severe and life-threatening for both the mother and child [8,10]. Therefore, screening for and, if indicated, treatment of PAVMs in all females with HHT considering pregnancy should be a priority [10]. Unfortunately, despite the potential exacerbation of epistaxis which may occur during pregnancy due to HHT, recommendations for the management of epistaxis before and during pregnancy are largely missing from the literature. 

This study was undertaken to evaluate female HHT patients’ understanding of HHT and its potential for increasing pregnancy risks, as well as to characterize their experiences with HHT during pregnancy and delivery, especially as it pertains to epistaxis. The findings were the impetus to deliver recommendations for management of HHT in pregnant women from an otorhinolaryngology perspective and with a focus on epistaxis. 

## 2. Materials and Methods

Fifty-five adult female patients, diagnosed with HHT, were recruited from the outpatient clinic of the Department of Otorhinolaryngology at the University Hospital Regensburg for this retrospective, questionnaire-based study. Inclusion criteria included a definitive diagnosis of HHT, as determined by the presence of at least 3 of 4 Curaçao criteria. Due to the fact that some of the participants had been diagnosed with HHT prior to publication of the Curaçao criteria in the year 2000 or had received their diagnoses external to our clinic, we independently confirmed all HHT diagnoses according to the currently accepted criteria as part of routine follow-up appointments within our day clinic. All participants had a history of at least 1 completed pregnancy and had undergone treatment for nasal telangiectasias in our tertiary referral center between 2019 and 2020. Participants were identified via review of clinical records and were initially contacted by telephone to inform them of the study. Two weeks later, they received a study packet containing additional informing regarding the study, an informed consent form and the 17-item questionnaire (see Appendix A for an English translation). The anonymous questionnaire and the informed consent form were sent back to the study center in 2 separate envelopes, thereby preventing association of questionnaire results with any given respondent. Completed questionnaires and informed consent forms were returned by 46 of the 55 potential respondents. Results from these participants were included in the study and formed the basis of all analyses. All study-related activities were approved by the institutional review board of the University of Regensburg on 13 May 2020 (file number 20-1844-101).

The 17-item questionnaire was divided into four sections: general pregnancy history, awareness of the disease, screening, and treatment of epistaxis. The survey was used to query both the severity of the disease before and during pregnancy, as well as the level of respondent knowledge concerning possible complications. Further, it was intended to assess how females with HHT experienced the disease during pregnancy, especially in terms of epistaxis. Data processing and analysis were performed using SPSS Statistics 25 (International Business Machines Corporation; Armonk, NY, USA). Results were evaluated using descriptive statistics, as appropriate. 

## 3. Results

### 3.1. General Pregnancy History

Forty-six out of fifty-five (84%) potential respondents returned the questionnaire assessing their personal experiences with HHT and pregnancy and were included in the study. Respondents’ ages, provided by 45 out of 46 participants, ranged from 37 to 82 years, with a mean age of 61.7 years (standard deviation (SD) = 11.2 years) at the time of the study. On average, the respondents reported 2.3 pregnancies (SD = 1.6 pregnancies; range = 9 pregnancies) each, resulting in a total of 108 births. Mean age at time of delivery was 27.1 years (SD = 5.3 years; range = 22 years). Two-thirds of the participants provided information on the type of delivery. Briefly, out of 77 deliveries described, vaginal delivery without epidural anesthesia was reported in 61 deliveries (75%), whereas vaginal delivery with epidural anesthesia was reported in 12 deliveries (15%). Finally, cesarean delivery under general anesthesia or epidural anesthesia was performed in four women each (5%). 

### 3.2. Awareness of the Disease

Approximately 84% percent of respondents reported having been diagnosed with HHT only after all pregnancies had been completed (Table 1). Four percent reported receiving the diagnosis after the first, but prior to the last, pregnancy and 11% reported receiving the diagnosis before the first pregnancy. Of significant clinical importance, of the 7 respondents reporting having received the diagnosis prior to the completion of their last pregnancy, 2 stated that having received the HHT diagnosis had influenced their family planning.

It is possible that diagnostic and counselling procedures have improved over time, such that younger respondents and those with more recent pregnancies might be more likely to report having been informed of their HHT diagnosis prior to the completion of all pregnancies. Results obtained after dividing the respondents into an older and younger group at the arbitrary age of 60 years, the median age of respondents for the dataset, do not suggest a meaningful effect of age on the timepoint of HHT diagnosis (see Table 1). Figure 1 and Figure 2 present dot plots indicating the reported time points of HHT diagnosis as a function of respondent age (Figure 1) and the number of years since the most recent pregnancy (Figure 2). Again, no trend is observable suggesting that younger respondents or more recent mothers are more likely to have received a HHT diagnosis prior to the completion of all pregnancies. 

### 3.3. Patient Knowledge and Awareness

At the time of the survey, 42 of 46 respondents (91.3%) reported being generally unaware that HHT is linked to potential complications during pregnancy, while 4 women (8.8%) reported being aware of potential complications. Of these, 2 reported learning of the risks prior to their second pregnancies and 2 learned of the risks after completion of all pregnancies. No respondent reported having had knowledge of the potential risks HHT holds for expectant mothers before onset of the first pregnancy. For respondents younger than 60 years of age, 3 of 22 reported knowing that HHT was associated with complications in pregnancy, while 19 were not aware of any possible disease-related complications. Of the 23 respondents older than 60 years, only one reported knowing about possible complications, while 22 reported not being aware of them. When asked who informed them of potential complications, family members and the treating gynecologist were listed. No respondent reported having been informed about potential complications by an otorhinolaryngologist.

### 3.4. Screening

Of the 46 total respondents, only 1 reported having received screening for the presence of PAVMs before her pregnancy. Accordingly, 98% of the female HHT patients surveyed did not receive any pre-pregnancy screening. One respondent indicated that she had received treatment to avoid complications due to nose bleeding. In this patient, a skin graft of the thigh was inserted into the nose (septal dermoplasty) to prevent nosebleeds during pregnancy and delivery.

Subgroup analysis revealed that, in the younger age group (younger than 60 years), only one out of 22 responders (4.5%) had received a further examination to clarify organ manifestations associated with HHT. In the older age group, no patient reported further examinations. 

### 3.5. Epistaxis and Treatment

Of the 46 patients who completed the survey, 28 (61%) reported suffering from nosebleeds during their pregnancies. Of these, 46% stated that the severity of the nosebleeds was similar to that of those prior to pregnancy, whereas 27% of respondents reported increased frequency and severity of epistaxis and 27% reported decreased frequency and severity of epistaxis during pregnancy. 

Eight of the 28 (29%) respondents affected by epistaxis during pregnancy sought out treatment during pregnancy, while the remaining majority (71%) reported not having needed treatment during pregnancy. For those having received medical treatment for epistaxis, treatment strategies included nasal packing, bipolar cautery, laser therapy and a combination of these measures. When asked whether they were able to self-treat their nasal bleeding, nearly three-quarters of those affected (20 of 28 respondents) answered “yes”, one woman (4%) answered with “no” and seven women (25%) said that there was no need for self-treatment at that time. Nasal packing, nasal cooling and nasal ointment were listed as strategies used for self-treatment, when required. Most respondents (91%) reported not experiencing epistaxis during delivery, while 2% of respondents did report nosebleeds and 7% could no longer remember. No participant reported nasal packing as having been necessary during delivery. 

Forty-three out of 46 participants answered a subset of questions addressing their desire and need for medical advice and counseling. Thirty-three percent of respondents reported that they would have found professional HHT counseling helpful, 30% said that counseling would not have been helpful and 37% answered the question with “I don’t know”. Of the 33% (14 participants) who would have found counselling helpful, 9 (64%) suffered from epistaxis during pregnancy, while the remaining 5 participants (36%) had not.

## 4. Discussion

### 4.1. Knowledge of Women with HHT about the Disease and Its Complications

A systematic inquiry into diagnostic time delay has suggested that patients receive a definitive diagnosis of HHT more than two decades after the onset of HHT-specific symptoms, on average [7]. Accordingly, family planning is often complete by the time of diagnosis. This finding is in accordance with the results of the retrospective cohort study presented here. We surveyed a total of 46 mothers with HHT who were treated for the disorder in our tertiary referral hospital specialized in the management of HHT. Taken together, over 84% of all respondents indicated that they had received diagnosis of HHT after family planning had been completed. In the majority of cases, pregnancy in females with HHT proceeds uneventfully. In the rare cases in which complications do occur, however, knowledge regarding the disease and its potential effects can play a major role.

A study by Shovlin et al. analyzing 262 pregnancies in 111 women with HHT showed that, in women experiencing a life-threatening event, prior awareness of HHT or PAVM diagnosis was associated with improved survival. In this cohort, the proportion of women not having received the diagnosis of HHT at the time of pregnancy was 74% [10]. 

In a descriptive case series of 244 pregnancies from Gussem et al., major complications were identified for 13% of women, none of whom had been screened or treated for PAVMs and CVMs prior to pregnancy. In our study, 42 women (91%) indicated that they were not aware of HHT being associated with complications during pregnancy either at the time of pregnancy or at the time of questioning. While international guidelines addressing this topic were recently published, the information does not yet appear to have become common knowledge among female HHT patients. It remains to be seen whether patient age significantly influences knowledge of HHT and pregnancy-related risk, though results of the present study suggest that it might not. None of the remaining four women (9%) who knew about the potential for complications during pregnancy received information before or during their first pregnancy. One patient (2%) was screened for PAVMs prior to the onset of her second pregnancy and another patient (2%) reported having received operative treatment (Saunders operation) to prevent nasal bleeding in preparation for pregnancy. 

### 4.2. Epistaxis during Pregnancy and Delivery in HHT Patients

The results of this study confirm previous reports suggesting both that epistaxis in HHT pregnancies is an important issue for those affected and that there was a general lack of education opportunities and assistance for women with HHT. Nearly two-thirds of the respondents (28/46, 61%) reported having suffered from epistaxis during their pregnancies. The vast majority (85%), however, did not know about their underlying condition at that time. Unfortunately, this is not surprising given the dearth of literature on this topic. In a large retrospective case series, 30.8% of patients reported that they had an increased frequency of epistaxis during pregnancy [11]. This finding is in line with the 27% of respondents reporting increased epistaxis frequency in our survey. To the best of our knowledge, this study is the first to evaluate the course of epistaxis in HHT pregnancies, as neither the recommendations for the management of HHT pregnancies listed in the publication by Shovlin et al., nor the current guidelines, address this issue [10,12]. 

In our survey, 8 (29%) of 28 respondents stated that their nosebleeds were so pronounced that they needed medical treatment, with treatment strategies including bipolar cautery and laser therapy, nasal packing as well as moisturizing and cooling of the nose. Within the scarce available literature on epistaxis in pregnant women, conservative treatments, such as intravenous tranexamic acid administration, nasal packing and bipolar cautery, are generally recommended as first-line measures. If conservative measures fail, surgical care under general anesthesia may need to be considered [13,14]. The question remains, however, whether these recommendations are generalizable to expectant mothers with HHT. 

### 4.3. Recommendations for Management of Females with HHT from an Otorhinolaryngology Perspective

Results of our survey confirm that there is a need for increased education and counseling efforts on the topic of HHT. Our intention was therefore to provide patients and treating physicians with a clear and short recommendation that could serve as a guide for the management of females with HHT before and during pregnancy. Considerations taken into account in preparation of the presented recommendations summarize information from the current guidelines and literature and are supplemented by us with information on the subject of epistaxis (see Figure 3). For further information on the subject of HHT and pregnancy in general, we highly recommend the current guidelines, which deal with HHT and pregnancy in detail [12].

The treatment of nasal HHT lesions can be divided into two categories: prevention and therapy. Special attention should be paid to bleeding prevention to avoid the stressful situation of acute, insatiable epistaxis, as well as the potential adverse effects of medication and general anesthesia for the expectant mothers.

As reinforced by our study participants, lubrication of the nasal mucosa with ointment, oils and saline is an established part of basic nasal care and has a preventive effect on nasal bleeding [15]. In addition, we recommend temporal nasal occlusion (tNO) by tape, producing a moist chamber and preventing nasal airflow [16]. Thanks to its simplicity and minimal adverse effects, we consider it convenient as a preventive measure in pregnant women with HHT. 

Another key element of prevention is laser treatment, which can be used to treat telangiectasias intranasally with great precision and minimal side effects for the surrounding mucosa [17]. Due to the chronic nature of the disease, laser therapy must be repeated on a regular basis and should, especially during pregnancy, be performed under local anesthesia. We recommend a Blue light laser (445 nm) technique, which is highly specific to blood vessels and HHT lesions, but also Nd:YAG laser or KTP laser treatment is considered suitable [15,17].

As recommended for local anesthetics during pregnancy in dental medicine, we recommend the application of lidocaine topically and articaine submucosally to anesthetize the nasal mucosa prior to treatment [18]. 

In order to minimize the need for emergency treatment of epistaxis in late-stage pregnancy, at delivery and in the first weeks postpartum, preventive laser treatments are recommended in the months prior to delivery. The frequency of the procedures can be based on the course of complaints documented in an “Osler-Calendar”, a tool we created for otorhinolaryngology specialists to review treatment strategies and for patients to enhance their comprehension of the disorder [19].

Bleeding telangiectasias can be treated with high-frequency alternating current, that is, bipolar radiofrequency. Again, treatment under local anesthesia is preferable and a detailed description of the procedure can be obtained elsewhere [15].

The efficacy of systemically administered tranexamic acid (TA) in HHT patients suffering from epistaxis has been established [20,21]. Although it is generally considered safe for intravenous administration, even in acute bleeding during pregnancy and during delivery, according to drug and medication information, it is contraindicated in early pregnancy and should only be used in case of vital threat during late pregnancy [22,23]. Furthermore, there are reports of an HHT-related increase in protein levels of coagulation factor VIII, which promotes thrombotic risk in these patients [24]. Therefore, the continuation of the drug cannot be recommended. In individual cases and for acute emergency treatment with TA, personalized risk-benefit considerations are required. A general recommendation for the use of TA in pregnant HHT patients cannot currently be provided given an inadequate evidence base.

Approximately 61% of women responding to our questionnaire reported being affected by epistaxis during pregnancy. Of these, 71% were able to treat themselves with nasal packings. It is generally accepted that the ability of HHT patients to apply nasal packings without assistance significantly improves quality of life. Thus, our tertiary care center considers training patients in this technique an integral part of the comprehensive care provided [25]. Anemia due to prolonged nasal bleeding can lead to severe complications for both infant and mother [26]. Thus, nasal self-packing is of particular importance for pregnant women with HHT. ENT doctors are generally not on hand at the time of delivery, nor are nasal packings suitable for HHT treatment commonly within reach. Therefore, instruction on self-treatment, as well as the provision of suitable medical nasal packing materials, is a critical part of our recommendations.

To prepare patients for the event of acute nosebleeds, we provide a graduated scheme for nasal self-packing. In mild-to-moderate bleeding, we prefer soft, absorbable sponges (e.g., polyurethane or gelatine), whereas extensive bleeding is best treated with pneumatic nasal packings (PNP). It is important that only tamponades made of smooth material, which is atraumatic for the vulnerable mucous membrane (for example, polyurethane or latex), be used.

In our questionnaire, 1 respondent reported having undergone an operation under general anesthesia in preparation for pregnancy (septal dermoplasty). In no instance was treatment under general anesthesia for acute nasal bleeding necessitated. Information in the literature on how often surgical treatment of epistaxis under general anesthesia is needed in pregnant HHT patients is currently lacking. If the bleeding of nasal lesions appears uncontrollable despite all of the above measures, the need for surgical treatment under general anesthesia is to be discussed with the treating gynecologists and anesthesiologists. Shovlin et al. suggested that, in the rare case of treatment under general anesthesia being required, a modified induction regimen with opiates should be used [10]. 

As a part of multidisciplinary management, counseling regarding the disease and its pregnancy-associated risks should be provided by several disciplines, including gynecology, otorhinolaryngology and internal medicine. The ENT physician has an important role in this constellation, as the HHT patient is seen by the otorhinolaryngologist prior to the onset of pregnancy most frequently. The following points are to be addressed during the interdisciplinary counseling of pregnant women with HHT and are based on existing guidelines and expert recommendations [10,12]: In the context of pregnancy-associated complications, HHT pregnancies are considered “high risk” if the women have untreated pulmonary AVMs or brain VMs or have not been screened. Women should be advised that there are red-flag symptoms, such as hemoptysis and sudden severe dyspnea. When these symptoms occur, immediate hospitalization is obligatory. In the presence of untreated PAVMs and/or CVMs, or in the absence of screening for PAVMs, multidisciplinary care in a tertiary center is recommended. In addition, presentation to a clinical geneticist should be advised when the following topics are of particular interest: family planning, inheritance and penetrance of HHT, prenatal diagnostic options, and postnatal genetic testing. 

Further aspects must be considered in the context of gynecological care. Namely, scheduling of the delivery must account for the known visceral manifestations, the family history and the individual medical history of the patient. Screening for visceral manifestations during pregnancy must be obtained in accordance with guidelines. For all pregnant women with HHT, antibiotic prophylaxis during delivery is recommended [10,12]. 

### 4.4. Future Research Directions

The findings of this study are limited in several aspects, each of which represents a direction for future research. First, the relatively long period of time between the participants´ pregnancies and the time of the survey increases the probability that participants recalled HHT-related details from past pregnancies inaccurately. Future work assessing the experiences of mothers with HHT more closely following childbirth is needed to better account for recall bias in study results. However, given that most women of childbearing age will not yet have received a reliable HHT diagnosis, the value of retrospective studies with longer intervals should not be underestimated. Further, as the older mothers included in our sample serve as valuable sources of HHT-related information to their children, we consider results describing their experience with and knowledge of HHT to be of significant clinical interest. Second, given the high median age of participants in our sample (60 years), it is possible that their experience is no longer reflective of that of younger mothers with HHT. For example, recent efforts in terms of early diagnosis of HHT and patient education suggest that a younger participant cohort might possess a greater level of knowledge and awareness on the topic than did our older cohort. Such an effect would serve to limit the generalizability of our findings. Importantly, though limited in terms of sample size, this study included respondents as young as 37 years and results did not reveal any effect of participant age. For example, results of our subgroup analysis indicated that reports from participants older than 60 years were similar to those of the younger participants. As such, time since the most recent pregnancy might not be as great a limiting factor as might initially be presumed. Additional studies are needed to systematically assess for any effect of age on a patient’s experience with HHT and pregnancy. Third, while difficult to avoid in the context of a single-center trial evaluating a rare disease such as HHT, with only 85,000 HHT patients in Europe, more females than males with different clinical presentations, the small sample size precluded advanced statistical evaluation [27,28]. Longer-term and multi-center study designs should be considered in future work to overcome these limitations and provide much-needed detail in this area. 

## 5. Conclusions

The recommendations for management of HHT in pregnant women from an otorhinolaryngology perspective, formulated as a consequence of our study results, are not intended to replace existing guidelines. Rather, these recommendations are designed to complement available guidelines and help close a gap in knowledge about epistaxis in pregnant women with HHT. Results of the present study indicate a significant need for appropriate counseling on the manifestations of HHT in pregnancy. To achieve this, increased awareness of this disease among health care professionals, in particular ENT doctors and emergency physicians, is critical. However, it is just as important that medical information on HHT be received by affected patients. Women with HHT must be educated about organ screening, medical treatment and self-treatment of epistaxis and possible pregnancy-related complications prior to pregnancy. The knowledge and implementation of recommendations for management of HHT pregnancies is indispensable in reducing the rare but potentially life-threatening complications of HHT during pregnancy. 

## Figures and Tables

**Figure 1 jcm-11-02178-f001:**
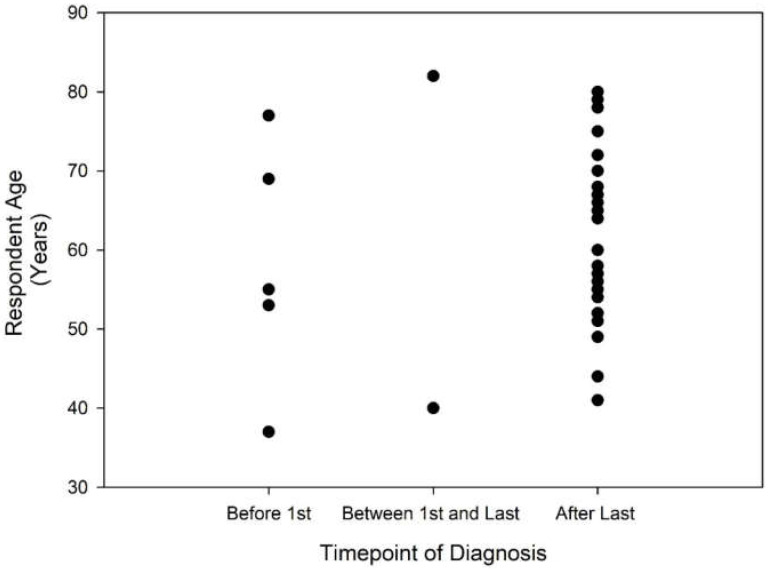
Respondent age as a function of the timepoint of HHT diagnosis.

**Figure 2 jcm-11-02178-f002:**
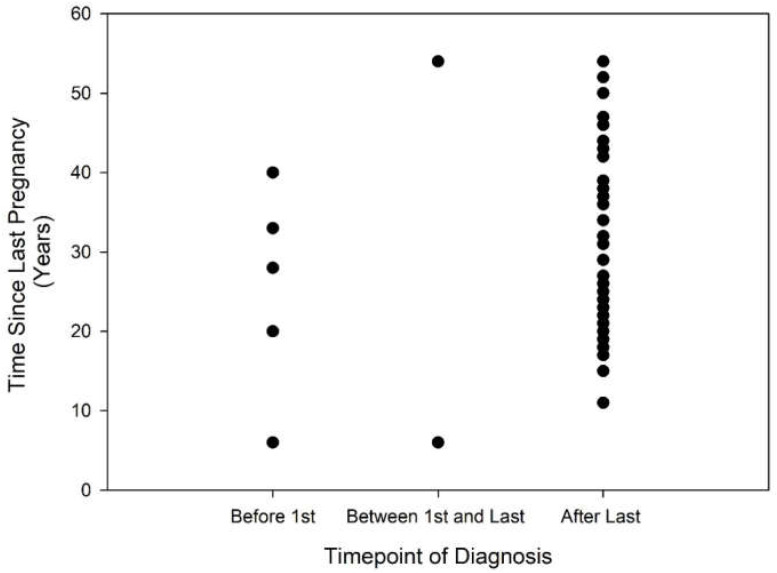
Time since last pregnancy (years) as a function of the timepoint of HHT diagnosis.

**Figure 3 jcm-11-02178-f003:**
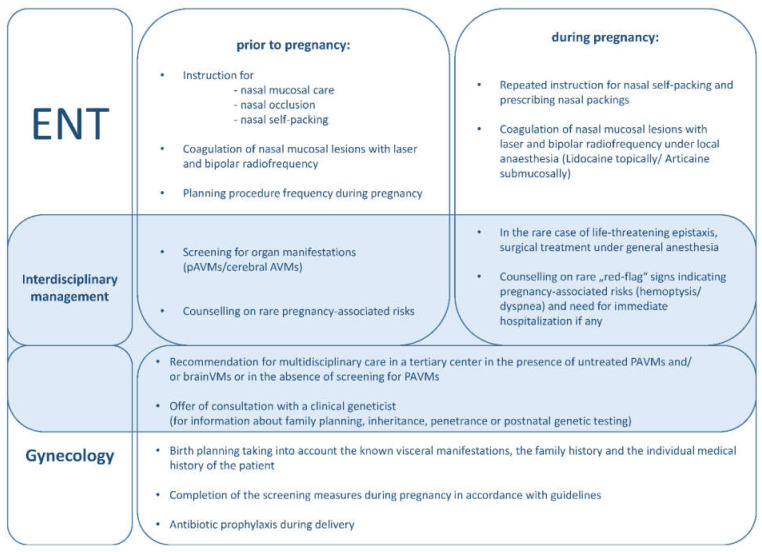
Recommendations for Management of HHT-related Epistaxis and Pregnancy from an Otorhinolaryngology Perspective. Items “Interdisciplinary management” and “Gynecology” in keeping with general advice with [10,12]. ENT = Ear, nose and throat physician; HHT = Hereditary hemorrhagic telangiectasia.

**Table 1 jcm-11-02178-t001:** Timing of HHT diagnosis relative to pregnancy (*n* = 45).

	All Respondents *n* = 45	Age at Survey < 60 Years	Age at Survey > 60 Years
Prior to first pregnancy	5 (11.1%)	3 (13.6%)	2 (8.6%)
Between first and last pregnancy	2 (4.4%)	1 (4.5%)	1 (4.3%)
After last pregnancy	38 (84.4%)	18 (81.8%)	20 (86.9%)

0 = completely unsatisfied; 10 = completely satisfied.

## Data Availability

The datasets used and analyzed during the current study are available from the corresponding author upon reasonable request.

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
