# Peer review of "HHT-Related Epistaxis and Pregnancy—A Retrospective Survey and Recommendations for Management from an Otorhinolaryngology Perspective"

_jcm, 2022, doi:10.3390/jcm11082178_

Round 1

Reviewer 1 Report

Introduction

This is a well-written section attracting readers' attention on the occasionally extremely long diagnostic delay of HHT and its implications on pregnancy and delivery.

Materials and Methods

Page 2, lines 78-80: Fifty-five adult female patients, diagnosed with HHT according to the Curaçao Criteria, were recruited. Please clarify their HHT status. Were they all definite HHT patients?

Page 2, lines 89-90: „Completed questionnaires and informed consent forms were returned by 46 of the 55 potential respondents”. This is a result, repeated in the Results section. Please omit this redundant information from the Materials and Methods section.

Results

Page 3, lines 117-118: „Approximately 84% percent of respondents reported having been diagnosed with HHT only after all pregnancies had been completed (Table 1) ”. How was HHT diagnosed before the establishment of Curacao criteria in 2000? Did you check the HHT diagnostic method(s) in patients' medical records? Were the patients diagnosed with HHT before 2000 reevaluated upon the Curacao criteria?

Page 3, lines 120-122: „Of significant clinical importance, of the 7 respondents reporting having received the diagnosis prior to the completion of their last pregnancy, 2 stated that having received the HHT diagnosis had influenced their family planning. ” How?

Discussion

The „Recommendations for management of females with HHT from an otorhinolaryngology perspective” is a well-written summary of the topic.

As the genetic screening for the family-specific mutation is available in several genetic laboratories throughout Europe, I would involve the role of clinical geneticists in family planning. Patients must be aware of the autosomal dominant inheritance and the age-related penetrance of HHT. Prenatal diagnosis is available, but it is not crucial as HHT is not a syndrome incompatible with life and a near-normal lifespan is available with the adequate screening and management of HHT related pathologies. Postnatally, genetic screening is available even in newborns, as a part of systematic family screening of HHT.

Figure 3, completed with the role of geneticists, should be transposed to its first reference in line 253.

Appendix1

Item No. 3.: „Has your illness influenced your family planning or desire to have children? ” Were HHT genetic tests performed before/during pregnancy? Did any of the participants require prenatal HHT diagnostic test for the fetus?

Item No. 13 and 14: these numeric rating scales should have been compared with mean epistaxis severity in the year preceding pregnancy.

Summary

Andorfer et al attempt to introduce the relatonship of a rare genetic disease and pregnancy. They present a good summary of ENT measures regarding pregnancy and delivery. Although the authors are aware of the limitations of this study (retrospective design, recall and diagnostic biases), these limitations greatly influence the adequacy of this study. Being aware of the the mean time of deliveries (27.1 years) and the mean age of participants (61.7 years), patients had to recall their more than 30-year-old memories! In other respects, this study reflects the patients' awareness of the effects of pregnancy on HHT status of the 1980s and 1990s. The diagnostic principles of HHT were not uniform prior to the introduction of Curacao criteria in 2000.

Author Response

Response to Reviewer 1 Comments

Point 1: Materials and Methods: Page 2, lines 78-80: Fifty-five adult female patients, diagnosed with HHT according to the Curaçao Criteria, were recruited. Please clarify their HHT status. Were they all definite HHT patients?

Response 1: Thank you for bringing this unclear wording to our attention. All patients included in the study from our HHT consultation at the Ear Nose and Throat department of the University clinic of Regensburg met 3 or 4 of 4 Curaçao criteria. Thus, they all had a definite diagnosis of HHT. The manuscript has been updated to clarify this point.

Point 2: Materials and Methods: Page 2, lines 89-90: „Completed questionnaires and informed consent forms were returned by 46 of the 55 potential respondents”. This is a result, repeated in the Results section. Please omit this redundant information from the Materials and Methods section.

Response 2: Thank you for noticing this repetition. The sentence has been deleted from the material and methods section.

Point 3: Results: Page 3, lines 117-118: „Approximately 84% percent of respondents reported having been diagnosed with HHT only after all pregnancies had been completed (Table 1) ”. How was HHT diagnosed before the establishment of Curacao criteria in 2000? Did you check the HHT diagnostic method(s) in patients' medical records? Were the patients diagnosed with HHT before 2000 reevaluated upon the Curacao criteria?

Response 3: Thank you for pointing out this ambiguity in our manuscript. All participants in our study had a definitive diagnosis of Osler disease (see above). This was checked on the basis of the patient medical records. Patients who were diagnosed at our clinic before the year 2000 were diagnosed according to the criteria valid at that time. The diagnosis of HHT before 2000 was established with two out of four manifestations: epistaxis, telangiectasia, a family history or well-documented visceral involvement (Plauchu H, de Chadarévian JP, Bideau A, Robert JM. Age-related clinical profile of hereditary hemorrhagic telangiectasia in an epidemiologically recruited population. Am J Med Genet. 1989;32(3):291-297. doi:10.1002/ajmg.1320320302). Some of the patients were diagnosed before the year 2000 or not at our clinic. As part of the regular presentation in our day clinic, a renewal of the diagnoses of all patients with HHT was carried out using the criteria that came into force in 2000.

The following paragraph was added to our manuscript in the material and methods section, from line 81: Due to the fact that some of the participants had been diagnosed with HHT prior to publication of the Curaçao criteria in the year 2000 or had received their diagnoses external to our clinic, we independently confirmed all HHT diagnoses according to the currently accepted criteria as part of routine follow-up appointments within our day clinic.

Point 4: Results: Page 3, lines 120-122: „Of significant clinical importance, of the 7 respondents reporting having received the diagnosis prior to the completion of their last pregnancy, 2 stated that having received the HHT diagnosis had influenced their family planning. ” How?

Response 4: That is a very interesting follow-up question that unfortunately was not assessed by our questionnaire. As such, we do not feel comfortable speculating on the manner. The implementation of qualitative research methods in a future study would certainly be appropriate. 

Point 5: Discussion: As the genetic screening for the family-specific mutation is available in several genetic laboratories throughout Europe, I would involve the role of clinical geneticists in family planning. Patients must be aware of the autosomal dominant inheritance and the age-related penetrance of HHT. Prenatal diagnosis is available, but it is not crucial as HHT is not a syndrome incompatible with life and a near-normal lifespan is available with the adequate screening and management of HHT related pathologies. Postnatally, genetic screening is available even in newborns, as a part of systematic family screening of HHT. Figure 3, completed with the role of geneticists, should be transposed to its first reference in line 253.

Response 5: Thank you for this comment. We have not yet highlighted this aspect in our manuscript. In Figure 3, the following aspect has been added in the section ” interdiciplinary management”: “Offer of consultation with a clinical geneticist (for information about family planning, inheritance, penetrance or postnatal genetic testing)”. The Figure 3 was moved to its first reference. In addition, the following sentence was added in the section „Recommendations for management of females with HHT from an otorhinolaryngology perspective, from line 346: „ In addition, presentation to a clinical geneticist should be advised when the following topics are of particular interest: family planning, inheritance and penetrance of HHT, prenatal diagnostic options, and postnatal genetic testing.“

Point 6: Appendix 1: Item No. 3.: „Has your illness influenced your family planning or desire to have children? ” Were HHT genetic tests performed before/during pregnancy? Did any of the participants require prenatal HHT diagnostic test for the fetus?

Response 6: Though of signifciant clinicial interest, this question was not adressed by our survey.

Point 7: Item No. 13 and 14: these numeric rating scales should have been compared with mean epistaxis severity in the year preceding pregnancy.

Response 7:. Unfortunately, we cannot repeat the survey due to the retrospective nature of the study. However, we agree that this suggestion should be headed in future prospective studies.

Reviewer 2 Report

The authors have presented their findings of a survey of patients with HHT and their experience with HHT using a retrospective study design. The authors acknowledge the mean/median age of the cohort may have led to responses that are not generalisable to the current HHT population.

Given the Curacao criteria were published in 2000, if not done, could the authors evaluate their survey results based on the year of publication of these criteria and/or following a sufficient evidence-based time frame for the clinical adoption of the Curacao criteria, in the event this impacts the author's research question and survey results? 

Author Response

Response to Reviewer 2 Comments

Point 1: I Given the Curaçao criteria were published in 2000, if not done, could the authors evaluate their survey results based on the year of publication of these criteria and/or following a sufficient evidence-based time frame for the clinical adoption of the Curaçao criteria, in the event this impacts the author's research question and survey results? 

Response 1: Thank you for bringing to our attention that our manuscript does not adequately address how the definitive diagnosis of HHT was made in our survey respondents. The section in question has been revised to clarify the timepoint of and basis for confirmation of the HHT diagnosis. Briefly, all participants in this study were determined to have confirmed HHT according to the Curaçao Criteria. Material and methods section: From line 81: “Inclusion criteria included a definitive diagnosis of HHT, as determined by the presence of at least 3 of 4 Curaçao criteria. Due to the fact that some of the participants had been diagnosed with HHT prior to publication of the Curaçao criteria in the year 2000 or had received their diagnoses external to our clinic, we independently confirmed all HHT diagnoses according to the currently accepted criteria as part of routine follow-up appointments within our day clinic.’’  
